# Cytotoxic, anti-inflammatory, antioxidant, and anti-glyoxalase-I evaluation of chelating substances: In silico and in vitro study

MohammedBashar Al-Qazzan[1,2]*, Qosay Al-Balas[3], Belal Alnajjar[1], Violet Kasabri[4]*, Yusuf Al-Hiari[4], Aref Zayed[3], Rema AlKhateeb[4], Mohammed Al-Akeedi[5]

1 Department of Pharmaceutical Sciences, Faculty of Pharmacy, Al-Ahliyya Amman University, Amman, Jordan, 2 College of Pharmacy, University of Bilad Alrafidain, Diyala, Iraq, 3 Department of Medicinal Chemistry and Pharmacognosy, Faculty of Pharmacy, Jordan University of Science & Technology, Irbid, Jordan, 4 School of Pharmacy, University of Jordan, Amman, Jordan, 5 College of Pharmacy, The University of Mashreq, Baghdad, Iraq

* alanim050@gmail.com (MBA); violetk70@gmail.com (VK)

## Abstract

### Introduction

Glyoxalase I is a crucial target in cancer treatment due to its involvement in detoxifying methylglyoxal. Suppressing the activity of Glyoxalase I has great potential for disrupting the pathways that allow cancer cells to survive, creating a new route for cancer treatment.

### Methodology

This study aims to investigate several substances as inhibitors of glyoxalase-I. After demonstrating their efficacy, additional inquiry will concentrate on assessing their cytotoxic and anti-inflammatory capabilities, yielding significant insights for prospective medicinal uses. The selection of these chemicals as potential Glyoxalase I inhibitors was based on their ability to bind metals, in crucial enzymes. These compounds consist of catechol, dihydroxy, trihydroxy benzene, and functional groups such hydroxamic acid, sulfur, carboxylic acid, amide, and 3-hydroxy-4-pyridone with the aim to suppress Glyoxalase I *in vitro*. Docking protocol was employed to examine the binding interactions with the active site. These auspicious compounds have been subjected to cytotoxicity assay testing employing Sulforhodamine B in eleven distinct cell lines, as well as anti-inflammatory and antioxidant evaluations.

### Results

Glyoxalase I inhibition revealed that 4-hydroxy-estradiol exhibited the highest efficacy with an $IC_{50}$ value of 0.226 µM. Trichostatin A demonstrated a significant anti-proliferation effect in colorectal cancer cells (CACO2, HCT116, SW620, HT29) with

**Data availability statement:** All relevant data are within the manuscript and its Supporting information files.

**Funding:** The author(s) received no specific funding for this work.

**Competing interests:** The authors have declared that no competing interests exist.

$IC_{50}$ values ranging from 14.0 µM to 27.0 µM. Furthermore, it exhibited substantial decreases in viability, ranging from 1.4 µM to 14.7 µM, in cancer cell cultures of skin (A375), lung (A549), prostate (PC3), breast (MCF7 and T47D), and cervical (HeLa). Except for 4-hydroxyestrdiol, the other phytochemicals demonstrated considerable selectivity in reducing the viability of cancer cells in monolayers of colorectal, cervical, mammary, lung, and skin tissues. In comparison to indomethacin and vitamin C, all of the studied natural compounds exhibited excellent anti-inflammatory properties in LPS-primed RAW 264.7 murine macrophages and had moderate antioxidant abilities comparable to ascorbic acid, with $IC_{50}$ values from 26.0 µM to 99.0 µM.

## Conclusion

Further assessment of molecular action mechanisms and structural betterments/ enhancements and/or derivatization of promising potent agents with antiglyoxylase – antiinflammation duality along with differential cytotoxicity and reductive capacities are advisably warranted with appropriately matched downstream in vivo validation modalities.

## Introduction

Cancer is a condition characterized by uncontrolled cellular proliferation that occurs in the human body due to the accumulation of genetic and epigenetic changes in normal cells [1]. Cancer is a significant global health issue, despite ongoing endeavors to address it [2]. Hence, it is crucial to discover innovative pharmaceuticals and study cancer behavior in order to uncover new targets for medication development. The potential of selectively inhibiting Glyoxalase I (Glo-I); as a therapeutic target, has been identified with promising outcomes [3–5]. The glyoxalase system carries out a detoxification function within living cells. It participates in the catalysis of GSH-dependent reactions by converting oxoaldehydes, such as methylglyoxal, into D-lactic acid, as illustrated in Fig 1 [6–8]. Methylglyoxal is a noxious byproduct of regular metabolism, generated during the glycolysis process [9]. The glyoxalase system consists of two enzymes, glyoxalase I (Glo-I) and glyoxalase II (Glo-II), along with some amount of glutathione (GSH) that acts as a catalyst. It has been suggested that inhibiting the glyoxalase pathway, particularly Glo-I, in cancer cells leads to the accumulation of toxic methylglyoxal, which in turn induces self-destruction of the cancer cells [10]. Advanced Glycation End Products (AGEs) play a major role in the development of diabetes, cardiovascular disease (CVD), and cancer. Glo-I is heavily involved in the body's defenses against glycation [11]. Inhibiting the Glo system has shown great promise as a potential therapy for treating and preventing tumors [12–14]. Increased levels of Glo-I, as a biomarker for tumors, have been linked to multidrug resistance in cancer chemotherapy [5,15,16]. Additionally, novel Glo inhibitors with a "zinc-binding feature" have been reported to possess anticancer properties, similar to anticancer metallo-drugs that inhibit cell proliferation through mitochondrial intrinsic apoptosis

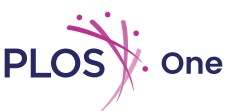

**Fig 1. The role of glyoxalase system in methylglyoxal detoxification.**

pathways [11,17–26]. Unequivocally nutritional Intervention of glyoxylase were underscored as prominent anti-Ageing strategies and glyoxalase I mimetics were principally considered as novel anti-glycation therapeutic agents [11,12,18–29].

Inflammation triggers the onset and progression of carcinogenesis. In the inflammatory tumor microenvironment, polarized inflammatory cells have the ability to release cytokines, chemokines, and prostaglandins. This secretion serves to

stimulate tumor development, angiogenesis, and metastasis [27,30]. Undoubtedly, non-steroidal anti-inflammatory drugs (NSAIDs) significantly decreased the likelihood of developing colon adenomas, breast, prostate, and lung malignancies [31]. Cerivastatin exhibits superior dual cytotoxicity and anti-inflammatory effects compared to the other 8 lipophilic statins. This is attributed to its ability to chelate with 3,5-dihydroxyheptanoic acid [32]. Emodin was discovered to suppress inflammation, carcinogenesis, and the advancement of cancer in an animal model of colitis-associated intestinal tumorigenesis [33]. In vitro studies have shown that new fluoroquinolones with a C7-C8 ethylene diamine bridge, which are bidentate chelators have potential antioxidant, anti-inflammatory, and cytotoxic properties in a wide range of cancer cell lines [34].

Furthermore, Oxidative Stress is strongly linked to neurological illnesses, chronic inflammation, metabolic problems, and the growth, survival, and migration of malignancies [35]. ROS can primarily contribute to the occurrence of genomic instability and proapoptogenesis thereafter. Chronic inflammation can be triggered by factors such as obesity, smoking, infections, and exposure to environmental pollutants [36]; IL-1 primarily contributes to the initial phases of cancer development and the generation of reactive oxygen and nitrogen species (ROS and RNS). Additionally, it was discovered that it promotes the growth of cancerous cells and the buildup of mutations, intensifying the ability of tumor cells to spread, triggering angiogenesis, and causing the release of substances that support the growth of blood vessels and the spread of cancer. Given Oxidative Stress and Glyoxalase Pathway in Cancer development and dissemination; the involvement of antioxidants in the management and pathology of cancer is yet to be delineated [11,12,18–29,37,38]. According to findings from translational medicine, antioxidants that fight cancer can come from both natural and artificial sources. The comprehensive understanding of the molecular and cellular mechanisms underlying tumor-promoting pro-oxidative inflammation is crucial for the advancement of chemotherapy and chemoprevention strategies [39–48].

After achieving successful inhibition, additional research has been undertaken on cell lines and more sophisticated testing methods, including the assessment of anti-inflammatory and anti-oxidant properties along with assessing antiglyoxylase qualities were conducted.

## Materials and methods

### Selection criteria

The compounds that were examined in this study were selected based on specific criteria, including their capacity to coordinate metal ions, particularly zinc (Zn) atoms, in various enzymes within our body. SAHA (Vorinostat), trichostatin A (TSA), and acetazolamide are drugs that have been identified as potent zinc coordinators, as stated in the literature. Cimetidine works as a ligand for iron and copper, while Tolcapone binds with magnesium [49–52]. Furthermore, it has been previously reported that chemical compounds containing a catechol moiety, dihydroxy and trihydroxy benzene, exhibit inhibition of the Glo-I enzyme [53,54]. Additionally, chemicals with functional groups capable of coordinating with zinc, such as hydroxamic acid, sulfur, carboxylic acid, amide, or 3-hydroxy-4-pyridone, as well as chemicals containing a ketol moiety, were anticipated to exhibit some level of action against our intended target. The efficacy of these drugs was evaluated in vitro against Glo-I. Several drugs have been demonstrated to exhibit activity against our specific target. The active compounds underwent docking using LibDock and CDocker algorithms to determine their scores and visualize their binding interaction with the active site.

### Glyoxalase-I (Glo-I) enzyme inhibition assay

Solvents and chemicals were procured from Sigma-Aldrich Co. and Acros (ThermoFisher Scientific, New Jersey, USA). The study used a human recombinant Glo-I enzyme (rhGlo-I) that was produced using E. coli and had an N-terminal Met and 6-His tag. The enzyme was obtained from R&D Systems® Corporation in the USA. Synthetic compounds were tested for their inhibitory effects on the enzyme using a 10 mM DMSO stock concentration in triplicates. The manufacturer's protocol from R & D Systems, Inc. in Minneapolis, MN, USA was followed for the in vitro experiments. The reconstituted Glo-I enzyme (0.5 mg/mL) was thawed after being stored at −70°C in sterile deionized water. The sample was observed

using a microplate reader (SpectraMax Plus by Molecular Devices, Hampton, NH 03842, USA) in a 96 well plate. The monitoring was done at a wavelength of 240 nm for 200 seconds at 37°C. The substrate mixture was freshly concocted with a concentration of 100 mM glutathione and 100 mM methylglyoxal (MG) in assay buffer containing 0.5 M sodium phosphate dibasic/monobasic. It was then left undisturbed at room temperature for 15 minutes -duration of. The selected compounds for investigation are subsequently produced in testing concentration ranges using a 0.1 M solution of sodium phosphate dibasic/monobasic with a pH range of 7–7.2. The blank contained an assay buffer with MG substrate. The test compounds were combined with human recombinant Glo-I (50 µM) in a 96-well plate to determine their kinetic behavior. The $IC_{50}$ values were then obtained by measuring the percentage of enzyme inhibition and comparing them to myricetin [11,18,19,21–26,54–58].

**Viability assays for antiproliferative capacities of test compounds**

**Sulforhodamine B (SRB) assay.** The cancer cell lines were cultured in DMEM supplemented with 10% FBS (Bio Whittaker, Verviers, Belgium), HEPES Buffer (10 mM), gentamicin (50 µg/mL), L-glutamine (100 µg/mL), streptomycin (100 mg/mL), penicillin (100 µg/mL), and 4-(2-hydroxyethyl)-1-piperazineethanesulfonic acid (HEPES) Buffer (Sigma, St. Luis, MO, USA). The Sulforhodamine B used was obtained from Santa Cruz Biotechnology, Inc. in Texas, USA. We obtained the following cell lines for cytotoxicity screening: Human PC3 prostate cancer cell line (ATCC® CRL-1435), A375 human skin cancer cell line (ATCC® CRL-1619), A549 lung cancer cell line (ATCC® CCL-185), Breast cancer cell line MCF7 (ATCC® HTB-22), T47D (ATCC® HTB-133), HeLa uterine cervix adenocarcinoma cell line (ATCC® CRM-CCL-2), PANC1 pancreatic cell line (ATCC® CRL-1469), and colorectal cancer cell lines HT-29 (ATCC® HTB-38), HCT116 (ATCC® CCL-247), SW480 (ATCC® CCL-228), SW620 (ATCC® CCL-227), and CACO2 (ATCC® HTB-37). Periodontal ligament fibroblasts (PDL) were utilized to assess specific cytotoxicity. Cancer cell lines that have survived offer several benefits. They provide a consistent and uninterrupted source of cells, so bypassing any ethical concerns related to the use of human tissue. Additionally, they are cost-effective [59]. The cytotoxicity levels were assessed using the Sulforhodamine B (SRB) colorimetric assay, provided by Santa Cruz Biotechnology, Inc. in Texas, USA. The Spectro Scan 80D UV-VIS spectrophotometer, manufactured by Sedico Ltd. in Nicosia, Cyprus, was employed for this cytotoxicity screening [60–65]. The cells were co-incubated with substances or a reference agent at various doses ranging from 5 to 200 µg/mL. Cisplatin, a potent and traditional anticancer drug that induces programmed cell death, was used as a reference agent at concentrations ranging from 1 to 200 µM for the purpose of comparison. The reduction in cell viability was analyzed by plotting Dose-response curves and expressing the data as a percentage of the control optical density. The $IC_{50}$ values were obtained using regression analysis. The anti-proliferative activities were determined as the mean $IC_{50}$ concentrations of the investigated compounds, along with the standard deviation (SD) (n = 12) [66–72].

**Anti-inflammatory (nitrite) determination *in vitro*.** The RAW 264.7 mouse macrophage cell line (ATCC® TIB-71) was cultivated in high glucose DMEM with 10% foetal bovine serum (FBS), 100 U/mL penicillin, 100 µg/mL streptomycin, and 100 µg/mL L-glutamate. The cells were incubated at 37°C in a humidified environment with 95% air and 5% CO2. Confluent macrophages ($2 \times 10^5$/well) were cultured with lipopolysaccharide (LPS; 20 µg/mL; Sigma, St. Luis, MO, USA) and indomethacin (25–200 µg/mL) as the positive control, along with various concentrations (5–200 µg/mL) of treatment compounds. The cells were incubated for 24 hours. 100 µL of Griess reagent were combined with 100 µL portions of cell culture medium and left to incubate at room temperature for 10 minutes. The absorbance at a wavelength of 550 nm was measured using a microplate reader, namely the Spectro Scan 80D UV-VIS spectrophotometer manufactured by Sedico Ltd. in Nicosia, Cyprus. The nitrite concentration was assessed by comparing it to a standard curve of sodium nitrite. A cytotoxicity protocol using SRB was conducted to assess the impact of the test substances under study on RAW 264.7 viability [70–72].

**DPPH free radical scavenger assay.** This approach relies on the reduction of the radicals, which causes a change in colour from oxidized purple to reduced yellow. Primarily, Diphenyl-2-picryl-hydrazyl (DPPH) is reduced in a solution of methanol (MeOH)

when a chemical that donates hydrogen is present. This reduction is due to the creation of the non-radical form DPPH-H. The alteration in colour can be precisely assessed by employing a spectrophotometer within the wavelength range of 515–520 nm. Unlike other assays for radical scavenging, the DPPH radical is stable and yields consistent spectroscopic data. A 0.2 mM DPPH solution was diluted with MeOH and then combined with treatment compounds and ascorbic acid in a 1:1 concentration ratio using a 96-well plate. This resulted in a final concentration range of 6.25–200 µg/mL for the test agents. The solution was then incubated for one hour, isolated from light. Subsequently, the alteration in absorbance at a wavelength of 517 nm was quantified using a microplate reader manufactured by Bio-Tek Instrument in the United States. Ascorbic acid served as a strong and traditional benchmark for comparing the effectiveness of radical scavenging agents. The DPPH radical scavenging activity inhibition was calculated using the following equation, where A denotes photometric absorbance: The formula to calculate the percentage difference between a control and a sample is: (A control − A sample)/ A control x 100% [71,72].

## Computational methods

The compounds that demonstrated efficacy against our target were analyzed using the LibDock and CDocker algorithms in Biovia® Discovery Studio 2022. Human Glo-I enzyme structural models were downloaded from the RCSB Protein Data Bank. The crystal form with the PDB ID: 3W0T was selected as the working model due to its resolution of 1.35 Å—the highest among the reported Glo-I structures—offering the most accurate and comprehensive atomic information. The downloaded protein, crystallized, was processed with the "Prepare Protein" tool in BIOVIA Discovery Studio. The tool was used for correcting and refining the structure by eliminating common crystallographic errors, Structural modifications included filling in missing parts, optimizing flexible regions, and fixing chemical properties to physiological pH. Finally, the structure was energy-minimized to eliminate steric collisions and optimize the overall geometry to verify that it was suitable for subsequent computational modeling and simulation work. the seven compound was established for prepare and generate ligand using the "Prepare Ligand" tool of BIOVIA Discovery Studio. Ligand conformations were optimized, protonation state at physiological pH corrected, 3D structure created and tautomers in the process. The elimination of duplicate or problematic entries was also performed. Ligands were minimized with the CHARMM force field and the smart minimizer to position them in a stable, low-energy form in preparation for docking studies [73]. Molecular docking studies were conducted to evaluate the binding pose and interaction energy of the prepared Glo-I protein structure with the selected ligand molecules. Docking was initially performed using the LibDock module of BIOVIA Discovery Studio based on a site-featured algorithm, which aligns ligands with protein interaction hot spots—polar and apolar regions in the binding site. Default parameters were applied, with modifications to docking performance (High Quality Protocol), conformation method (Best), and minimization (Do Not Minimize). Predicted poses were assessed based on interaction energies and significant contacts with active site residues. For further validation of binding interactions, docking was also carried out with CDocker protocol using CHARMM force field, charging ligand partial charges via the Momany-Rone method. The generated poses were visually inspected to determine how well they fit into the active site and the quality of their interactions with important residues,in support of further consideration of their potential activity.

## Statistical analysis

The values were presented as mean ± SD of 12 independent experiments (per each design) and were determined using GraphPad Prism software [version 8.0 for Windows; GraphPad software, San Diego, CA, USA].

## Results and discussion

### Glo-I active compounds

The active compounds targeting the Glo-I enzymes were classified into two groups: highly active compounds and compounds with intermediate activity, based on their in vitro results as presented in Table 1 and illustrated in Fig 2. The

**Table 1. The inhibitory activity of the most active compounds against Glo-I enzyme with less than 50 μM. Their supportive corresponding docking values on two different protocols are displayed.**

| Name | IC$_{50}$ μM ± SD | LibDock score | - CDocker interaction energy |
|------|-------------------|---------------|------------------------------|
| #1<br>4-hydroxy-estradiol | 0.226 ± 26.43 | 95.88 | 32.87 |
| #2<br>Shikonin | 8.1 ± 29.25 | 109.41 | 42.89 |
| #3<br>Tolcapone | 9.6 ± 32.97 | 97.14 | 40.82 |
| #4<br>Quinalizarin | 33.4 ± 12.64 | 101.92 | 31.29 |
| #5<br>Hispidine | 36.4 ± 16.14 | 94.59 | 34.48 |
| #6<br>Trichostatin A | 43.5 ± 15.61 | 109.18 | 25.04 |
| #7<br>Calceolarioside A | 44.3 ± 18.64 | 147.64 | 58.94 |
| Positive control<br>(Myricetin) | 3.6 ± 36.28 | 123.69 | 56.07 |

compounds with an IC$_{50}$ value below 10μM are considered highly active, whereas those with values between 10 μM and 50 μM are classified as intermediate.

4-hydroxyestradiol is a naturally occurring catechol estrogen that is produced as a minor metabolite of estradiol. It has estrogenic properties that closely resemble other hydroxylated estrogen metabolites, including 2-hydroxyestradiol, 16-hydroxyestrone estriol (16-hydroxyestradiol), and 4-hydroxyestrone [74]. Furthermore, it has been identified as a highly effective inhibitor of the Glo-I enzyme, displaying an activity level of 0.22 μM. Consequently, it holds promise as a prospective candidate for use as an anticancer agent. Fig 3 illustrates the binding pattern of this chemical. As anticipated, the catechol moiety engages in interactions with the zinc atom located in the active site, whereas the hydroxyl group at position 17 interacts with the amino acid Met 157 at the entrance of the active site. The hydrophobic structure of the steroid molecule engages in hydrophobic interactions with the non-polar amino acids at the active site, specifically Leu69 and Phe62.

Shikonin and its derivatives are obtained from traditional medicinal plant species belonging to the Boraginaceae family. They were frequently employed in ancient medical practices to address a wide range of ailments, including but not limited to cancer, blood clot prevention, protection of nerve cells, diabetes management, viral infection control, and reduction of inflammation [75]. Additionally, it demonstrates strong inhibitory effects on the Glo-I enzyme, with an activity level of 8.1 μM, suggesting its potential as an anticancer medication. Fig 4 reveals an unexpected interaction between the terminal aliphatic OH group and the zinc atom, while the hydrophobic arm occupies the hydrophobic pocket. The keto moiety, known for its ability to coordinate with zinc, is unable to fulfil its function due to being tightly confined within the hydrophobic pocket, far from the zinc atom. The hydrophobic arm interacts with the hydrophobic amino acids located near the entrance, as anticipated. Also, it augments the potency of binding at the active site by forming hydrogen bonds with Cys 60.

Tolcapone functions as a catechol-O-methyltransferase (COMT) inhibitor, which is a crucial enzyme involved in the metabolic pathway of levodopa. It is used as a supplement to levodopa/carbidopa treatment in the control of Parkinson's disease [76]. With a potency of 9.6 μM against the Glo-I enzyme, it demonstrates strong inhibitory activity, indicating its effectiveness as an enzyme blocker and its potential as an anticancer medicine. Fig 5 illustrates the binding pattern of this chemical. The ketone and nitro groups directly bind to the zinc atom in the active site, as demonstrated. Furthermore, the

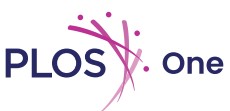

**Fig 2. Active compounds identified as glyoxalase-I inhibitors.**

nitro group engages in hydrogen bond interactions with the amino acid Thr101 near the entrance of the active area. The benzene rings engage in hydrophobic interactions with non-polar amino acids in the active site, namely Leu 69, Phe 62, and Phe162.

Trichostatin A (TSA), a fungal antibiotic derived from Streptomyces, functions as a potent inhibitor of mammalian histone deacetylases, particularly the class I and II families of enzymes. TSA has been demonstrated to elicit an inhibitory

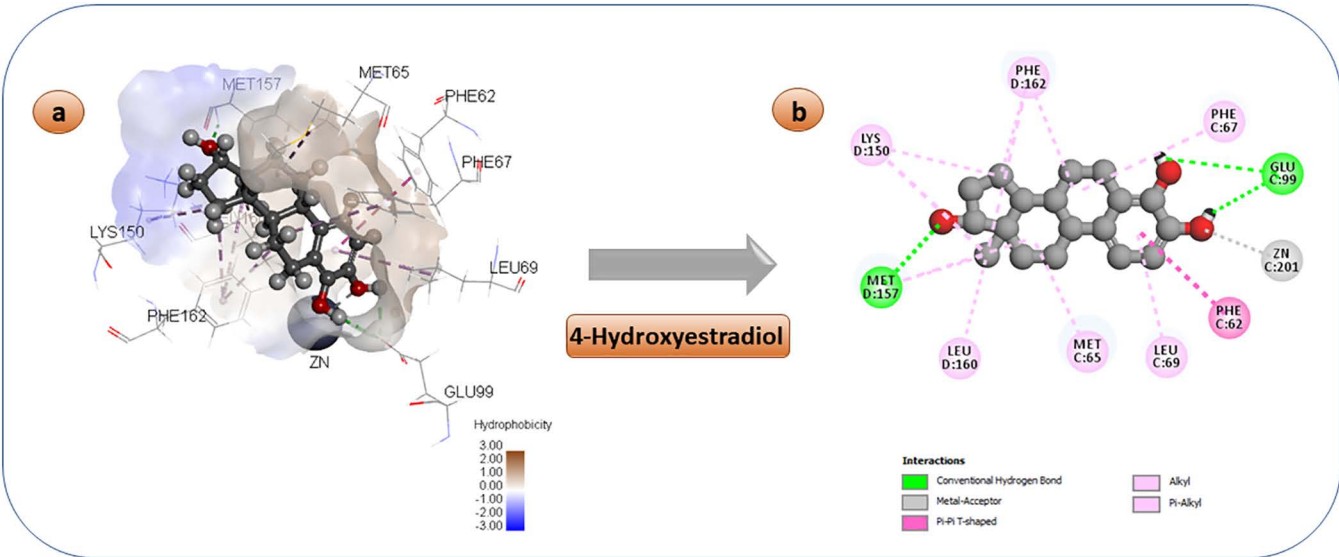

**Fig 3. 3D and 2D pose of the docked 4-hydroxyestrdiol diagram displaying a variety of interactions with enzyme's active region.**

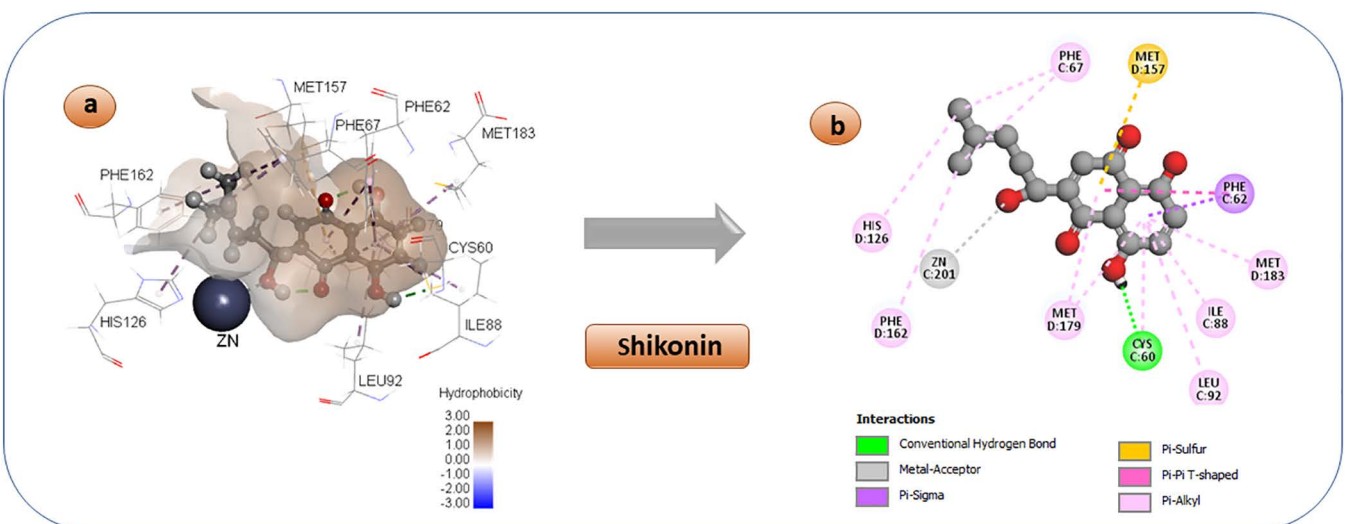

**Fig 4. 3D and 2D pose of the docked Shikonin diagram displaying a variety of interactions with enzyme's active region.**

response in the growth of certain types of cancers [77]. The presence of a hydroxamic acid group as a zinc coordinator in the compound may account for its moderate activity (43.5 µM) against the Glo-I enzyme. Calceolarioside A, a phenyl-propanoid glycoside previously detected in multiple Calceolaria species. It exhibits antinociceptive effects and possesses anti-inflammatory properties [78]. Furthermore, it exhibited moderate inhibition against Glo-I with an $IC_{50}$ value of 44.3µM, which is believed to be attributed to the presence of a catechol moiety. Quinalizarin is a hydroxy-9, 10-anthraquinone derivative that belongs to the family of anthracycline anticancer medications. It also acts as a protein kinase inhibitor [79]. The presence of a ketol moiety in the compound is likely responsible for its modest activity (33.4 µM) as it is expected to bind to zinc. Hispidin (HIP) is widely present in edible mushrooms, and several studies investigating its potential

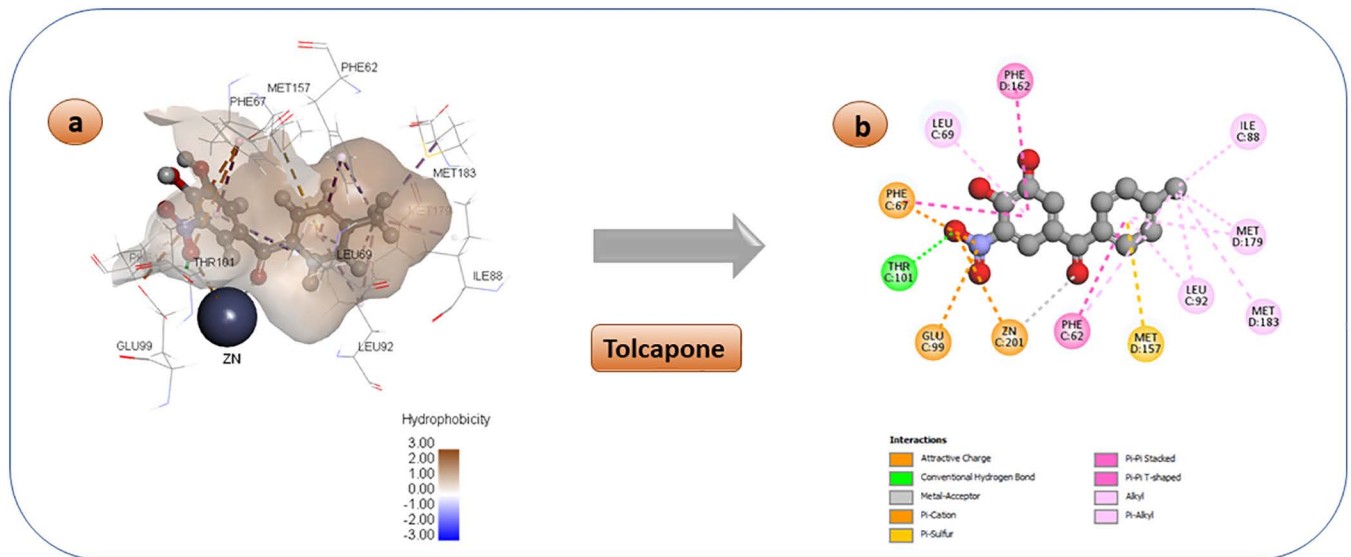

**Fig 5. 3D and 2D pose of the docked Tolcapone diagram displaying a variety of interactions with enzyme's active region.**

pharmacological advantages have been published [80,81]. Prior studies have demonstrated that HIP exhibits antioxidant capabilities, as well as anti-inflammatory, anticancer, and antiallergic effects, among other benefits. Furthermore, it exhibited moderate activity against Glo-I, with an $IC_{50}$ value of (36.4 µM), which is believed to be associated with the catechol component [82–86].

## Inactive compounds

The inactive compounds were categorized into various groups based on their functional groups, as seen in Fig 6. The first group in box A consists of a hydroxyl moiety that is linked to either an aromatic ring, a monocyclic conjugated ring, or a non-aromatic ring. The second group in box B consists of dihydroxy groups that are distinct from both catechol and trihydroxy groups. The third group in box C consists of catechol compounds that have either one or two carboxylic acid derivatives or amine carboxylic acid derivatives, with varying molecular weights. Box D has a methoxy group that is generated by methylating catechol groups. Furthermore, Box F has a hydroxamic acid group, which likewise serves as zinc coordinators. Box G has a nitrile functional group that serves as a zinc coordinator. Box H comprises several zinc coordination functional groups.

In details, the chemical moieties vary in size, ranging from small to large. The inactive compounds were categorized according to their size. Small-sized compounds, such as Tropolon, 3-Hydroxy-1,2-dimethyl-4(1h)-pyridone, Chloric acid, and Dimethylglyoxime, may not be able to enter the active site due to their high polarity. Even if they do manage to enter, they will not be able to fully inhibit the activity. Conversely, bigger molecules such as cynarine and 3,4-diaffeoylquinic acid are anticipated to have reduced activity since their size exceeds that of the active site. Consequently, molecules are unable to gain access to the active site and engage with it effectively.

It was anticipated that highly ionized compounds, such as Benserazide, Higenamine, and n,n-bis (alpha-cyano-4-dimethylaminobenzyl)ethylenediamine, would be ineffective because the ionizable amine group present in them is repelled by positively ionized amino acids near the entrance of the active site. Polar molecules such as caftaric acid, chrosmic acid, and withaferine A may face difficulty in reaching the active site because of the significant solvation penalty.



**Fig 6. Classification of inactive compounds.**

The selection of Terameprocol was based on its resemblance to the highly potent molecule (4,4'-((2R,3S)-2,3-dimethylbutane-1,4-diyl)bis(benzene-1,2-diol)), as described in a prior publication with an $IC_{50}$ value of 36.5nM. Terameprocol is rendered inactive due to the substitution of the catechol moiety with a methoxy group. By methylating catechol to form methoxy, the compounds were rendered inactive. This provides compelling evidence that the catechol moiety plays a crucial function in binding to the zinc atom [87].

SAHA and Zileuton exhibit potent zinc coordination via a hydroxamic acid group, but regrettably, they do not demonstrate efficacy against Glo-I. This can be ascribed to the compounds' incapacity to align with the geometric structure of

the active site. The compounds Gallocatechin, (+)Catechin, and (-)Epicatechin are inactive, possibly due to the presence of a chiral center in their molecules. This chiral center hinders appropriate binding, as previously demonstrated by our research group [21,88]. Cimetidine was selected because to its inclusion of the imidazole ring, which is recognized for its capacity to bind zinc, similar to the histidine amino acid. The imidazole ring appears to be incapable of coordinating with the zinc atom. The reduction of activity in Entacapone compared to Tolcapone (active) can be attributed to the presence of cyano and the substitution of toluene with diethyl tertiary amide. This loss of activity may be explained by the adoption of an inappropriate geometry [75].

Box H contains known chemicals capable of coordinating with zinc, such as acetazolamide, a carbonic anhydrase inhibitor (CAI). The compound contains sulphonamide functionality, which forms a bond with the zinc atom located within the active site of (CAI). However, it did not exhibit any action against Glo-I [89]. This suggests that although these medicines possess a zinc coordinating functional group, they exhibit selectivity towards their intended target and are incapable of inhibiting Glo-I action.

### Glyoxalase I (Glo-I) inhibition as novel and successful molecular mechanism of cytotoxicity in vitro

Tables 2 and 3 demonstrate that doxorubicin exhibited significant inhibitory effects on cell proliferation, with $IC_{50}$ values ranging from 0.006 µM to less than 7.2 µM, in cancerous monolayers of gastrointestinal (colorectal and pancreatic), skin, lung, prostate, cervical, and mammary tissues. The SRB bioassay was employed to evaluate the effects of cisplatin on the viability of cancer monolayers. Cisplatin demonstrated significant dose-dependent reductions in viability, with $IC_{50}$ values ranging from 1.9 to 33 µM. Remarkably similar to doxorubicin (Table 2), all relevant selective lack of cytotoxicities were evident in normal PDL fibroblasts by 72h incubations of all phytoprinciples, further supportive of their physiologically regulated efficacies [89]. Basically, these effects were in contrast to cisplatin's lack of selective cytotoxicity. Impressively Table 2 shows that trichostatin A, along with other phytoprinciples, significantly inhibited the development of colorectal and pancreatic cancer cells. Trichostatin A demonstrated a strong inhibitory impact with $IC_{50}$ values of ≤27 µM after 72 hours of incubation. Phytomedicine-derived bioactive compounds have demonstrated highly effective viability reduction capabilities (with $IC_{50}$ values ≤100 µM) in colorectal cancer cell lines CACO2, SW480, HCT116, SW620, and HT29. However,

**Table 2. Cytotoxicity (as of %Control) IC$_{50}$ value (mean ±SD in µM) of the tested compounds vs. Positive controls Cisplatin and doxorubicin against gastrointestinal and colorectal malignancies' adherent monolayers.**

| Treatment | CACO2 | SW480 | HCT116 | SW620 | HT29 | PANC1 | PDL fibroblasts |
|---|---|---|---|---|---|---|---|
| **4-Hydroxyestradol** | 106.58±4.77 | 109.45±6.86 | 124.3±7.8 | 142.29±16.5 | 118.35±4.17 | 66.9±2.69 | NI*** |
| **Trichostatin** | 27.05±1.28 | 44.34±4.67 | 14±2.5 | 19.5± | 27±0.85 | 95.8±8.2 | NI*** |
| **Calceolarioside A** | NT** | NT** | NT** | NT** | NT** | NT** | NI*** |
| **Quinalizarin** | 108.55±9.26 | 117.75±3.04 | 72.2±5.7 | 40.68±2.21 | 143.85±15.6 | 127.42±2.52 | NI*** |
| **Shikonin** | 43.82±4.97 | 41.63±6.37 | 108.2±2.6 | 60.27±2.07 | 142.26±0.76 | 80.01±2.11 | NI*** |
| **Telcapone** | 94.8±6.65 | 164.95±6.58 | 86.2±6.6 | 105.4±8.35 | 63.28±1.03 | 119.07±9.28 | NI*** |
| **Hispidin** | 74.3±1.13 | 34.6±3.11 | 198.5±12.2 | 34.26±6.52 | 82.75±0.78 | 129.8±2.08 | NI*** |
| **MYRICETIN** | 106.5±0.6 | 88.93±14.11 | 44.71±2.14 | 42.75±0.78 | 32.65±1.06 | 197.64±43 | NI*** |
| **Cisplatin** | 4.88±0.59 | 2.6±0.14 | 1.92±0.14 | 10±0.6 | 3.35±0.21 | 11.82±0.25 | 5.6±0.53 |
| **Doxorubicin** | 0.92±0.03 | 0.01±0.00 | 0.92±0.03 | 0.65±0.03 | 0.006±0.000 | 7.17±1.07 | NI*** |

**Notes:** Results are mean±SD (n=12). IC$_{50}$ values (concentration at which 50% inhibition of cell proliferation took place in comparison to non-induced basal 72h incubations) were calculated within 0.1–200 µM range. Lack of cytotoxicity of the same panel of tested compounds against normal PDL (periodontal ligament) fibroblasts vs. cisplatin and doxorubicin.

**NT**:**Not tested.

**NI**:*** Non-inhibitory within testing dose range"'.

**Table 3. Cytotoxicity (as of %Control) IC$_{50}$ value (mean ±SD in µM) of the tested compounds vs. positive controls Cisplatin and doxorubicin against other malignancy' cell lines adherent monolayers.**

| Treatment | A375 skin melanoma | A549 lung malignancy | PC3 prostate malignancy | MCF7 mammary malignancy | T47D mammary malignancy | Hela cervix malignancy |
|---|---|---|---|---|---|---|
| 4-Hydroxyestradol | 47.1±1.9 | 15.15±1.34 | 82.25±8.56 | 139.8±1.13 | 51.53±5.2 | 29.75±4.74 |
| Trichostatin | 5.6±0.6 | 2.0±0.21 | 47.5±3.96 | 2.25±0.49 | 14.7±0.28 | 1.4±0.2 |
| Calceolarioside A | NT** | NT** | NT** | 899.25±179.25 | 45.65±3.46 | 207.6±34.78 |
| Quinalizarin | 48.6±1.8 | 7.9±0.71 | 143.08±0.18 | 33.25±0.78 | 37.72±4.78 | 85.51±0.44 |
| Shikonin | 56.5±6.9 | 54±5.8 | 176.22±1.25 | 37.75±3.89 | 30.91±3.66 | 13.87±1.91 |
| Telcapone | 64.1±1.1 | 61.96±2.21 | 256.68±37.11 | 25.56±1.49 | 26.85±0.35 | 39.35±0.35 |
| Hispidin | 98.9±9 | 26.02±5.22 | 138.81±13.58 | 384.15±7.57 | 52.63±0.69 | 122.45±1.34 |
| MYRICETIN | 51.7±4.1 | 27.46±4.47 | 65.4±1.84 | 73.05±0.78 | 29.8±2.6 | 35.8±4.4 |
| Cisplatin | 15.7±0.6 | 33.03±2.49 | 6.27±0.95 | 12.8±1.98 | 12.15±0.35 | 22.75±1.2 |
| Doxorubicin | 0.28±0.14 | 0.28±0.00 | 0.13±0.01 | 4.81±0.21 | 0.02±0.00 | 5.56±0.08 |

**Notes:** Results are mean±SD (n=12). IC$_{50}$ values (concentration at which 50% inhibition of cell proliferation took place in comparison to non-induced basal 72h incubations) were calculated within 0.1–200 µM range. NT:**Not tested.

the investigated plant chemicals showed significantly lower levels of cytotoxicity in PANC1 pancreatic cancer adherent monolayers.

Table 3 demonstrates that Trichostatin exhibited significantly higher cytotoxicity affinities, with IC$_{50}$ values below 10 µM, compared to cisplatin, and similar to doxorubicin, in skin melanoma, lung carcinoma, mammary, and cervix adeno-carcinomas. The remaining phytoprinciples exhibited significantly moderate antiproliferative effects (with IC$_{50}$ values <50 µM) in cancer cell lines including cutaneous A375, mammary T47D and MCF7, and cervical Hela. The prostate PC3 72h incubations showed the lowest effectiveness in reducing viability when using repurposed phytoprinciples. However, the most significant effectiveness, with IC$_{50}$ values less than 25 µM compared to cisplatin's 33 µM, was observed in adenocar-cinoma monolayers.

## Glyoxalase I (Glo-I) inhibitors as novel and successful dual antiinflammation-cytotoxicity natural agents

Table 4 presents the significantly moderate abilities of phytoprinciples to scavenge DPPH radicals and act as antioxidants, compared to the capabilities of ascorbic acid (with IC$_{50}$ values ranging from 25 µM to 100 µM). Furthermore, all naturally occurring substances exhibited significant anti-inflammatory effects that were comparable to those of the nonsteroidal anti-inflammatory drug indomethacin, as shown in the same table. All IC$_{50}$ values were maintained at levels below 10 µM, which is a notable achievement. Myricetin had a remarkably strong ability to reduce inflammation, with an IC$_{50}$ value of 50nM. None of the evaluated medications' efficacies were significantly correlated with reductions in viability as validated by SRB [90–97].

It is important to note that, except for Trichostatin A, the most effective compounds against Glo-I, such as 4-hydroxy-Estradiol, Shikonin, and Tolcapone, have similar structural characteristics. They all possess at least one free Zn chelating group, which includes catechol groups or betahydroxy ketone. The exceptional inhibitory effect of Myricetin against Glo-I can be attributed to the presence of two Zn chelator groups. The instructions are sufficiently clear for most participants to follow by avoiding any disturbance with the proximity of Zn chelation. This was further confirmed and shown by the presence of trans-isomerism, which enforces a semi-planar/linear structure and, as a result, enhances activity. The mild cytotoxic activity of Glo-I active hits can be largely attributed to lipophilicity. There is ample evidence to support the fact that lipophilic substances readily permeate the cell membrane of cancer cells. The hydrophilic nature of our hits' structure accounts for their moderate to low cytotoxic activity.

**Table 4.** $IC_{50}$ values (mean ±SD in μM) of *in vitro* DPPH-radical scavenging properties vs. ascorbic acid and antiinflammation propensities vs. indomethacin of the tested compounds.

| Treatment | DPPH- $IC_{50}$ value μM | iNOS- $IC_{50}$ value μM |
|---|---|---|
| **4-Hydroxyestradol** | 506.35±102 | 0.25±0.04 |
| **Trichostatin** | 99.33±15.45 | 1.69±0.12 |
| **Calceolarioside A** | NT** | NT** |
| **Quinalizarin** | 579.95±26.52 | 1.85±0.34 |
| **Shikonin** | 34.37±5.04 | 0.27±0.05 |
| **Telcapone** | 92.1±2.4 | 1.19±0.17 |
| **Hispidin** | 26.26±0.65 | 3.47±0.49 |
| **MYRICETIN** | 70.73±11.64 | 0.07±0.01 |
| **Reference Drug** | Ascorbic Acid 0.09±0.01 μM | Indomethacin 0.02±0.00μM |

Notes: Results are mean±SD (n=12). $IC_{50}$ values (μM) (concentration at which 50% inhibition of DPPH reduction in comparison to non-induced basal incubations) were calculated within testing dose range. Lack of cytotoxicity of the same panel of tested compounds against LPS-primed inflammation in RAW macrophages vs. indomethacin. NT: not tested.

Table 2 demonstrates that the majority of active hits were mostly directed towards colorectal cell lines. This can be attributed to the overexpression of Glo-I in colorectal cancer (CRC), as supported by literature [98]. Therefore, this strongly confirm Glo-I as the primary target in CRC for cancer treatment. However, Table 3 displays that the majority of active hits display significantly modest abilities to inhibit the growth of cancer cells in skin A375, mammary T47D and MCF7, and cervical Hela. These findings indicate the involvement of Glo-I in these cells, along with other possible targets. Interestingly, the remarkable cytotoxicity of Trichostatin A against nearly all examined cell lines clearly indicates the significant significance of the Hydroxamic acid group, revealing additional targets such as Histone deacetylase (HDAC) receptors. The exceptional capacity of Trichostatin A to inhibit many enzymes, including Glo-I-HDAC, can account for its significant antiproliferative activity. This effect is likely due to its linear and lipophilic properties. Ultimately, this study reveals naturally occurring substances that inhibit Glo-I and have the potential to act against cancer cells.

**Speculative molecular action mechanisms of antitumor -antiinflammation cross talks biology.** To Speculatively elucidate the intricate linkage of inflammation- tumerogenesis relationship; naturally occurring methyl analogs of mushrooms' antcins had substantial cytotoxicities attributed to their ability to modulate signaling cascades in apoptotic pathways combined with antiiflammation capacities [99]. Reportedly Cathepsins are cysteine proteases that can be involved in regulable fundamental mechanisms as apoptosis, pyroptosis, ferroptosis, necroptosis, and autophagic cell death [100]. As they are cardiometabolic adiposity biomarker expressed in white adipose tissue; mostly elevated circulating concentrations of cathepsins can be strongly and independently associated with metabolic syndrome in overweight and obese adults [101]. Also, they are substantially involved in osteolysis, immunomodulation and apoptosis with a significant role in metastatic pancreatic cancer. In particular, targeting cathepsin subtypes was found promising in reducing tumor progression migration and invasion, further enhancing the efficacy of chemotherapeutic agents in preclinical models [102]. Recently antiproliferative α-fluorocinnamate-based cysteine protease inhibitors were principally proven as highly effective candidates targeting cathepsins in markedly reducing pancreatic cancer cell proliferation [102,103]. Interestingly chemerin, Defensin 1, and TNFα were underscored as potential biomarkers in the early diagnosis of colorectal cancer [104], tubulins can also serve the same purpose as their inhibitors being classified as anticancer agents [105]. Survivin can be an antiapoptotic peptide inhibiting caspases −3 and −7 via activation of intrinsic apoptogenic pathway [106] closely linked to antitumor immunity (Jazayeri et al., 2020) thereby underscoring rapid noninvasive detection of bladder cancer [107]. Moreover, In-depth, molecular action mechanism of cisplatin cardiotoxicity can afford translational evidence into cardioprotective effectiveness of safe antiproliferation compounds [108]. Substantial changes

to levels of proinflammatory IL-1β, −6 and antiinflammtory IL10 and SIRT1 can be recognized in their cross-talk of mechanistic immunomodulation [109,110].

**Concluding remarks and future directives.** To summarize, a range of compounds were tested for their impact on Glo-I enzyme activity. The compounds that exhibited inhibitory action were then assessed for their effects on cytotoxicity, anti-inflammatory properties, and antioxidant capabilities. As a result of these assessments, 4-hydroxyestridiol was identified as a potent inhibitor against Glo-I, with an $IC_{50}$ value of 0.226 μM. Unequivocally unlike undifferential cytotoxicity of cisplatin, the spectrum of safety of all tested agents in periodontal ligament fibroblasts PDL-based 72h incubations and RAW 264.7 macrophages were reportedly signified. Trichostatin exhibited notable inhibitory effects on the growth of cancer cells, leading to considerable reductions in cell viability in multiple cancer cell cultures. Nevertheless, tumour samples obtained from the prostate and pancreas exhibited remarkable resistance. Docking investigations are conducted to demonstrate the binding interactions that occur at the active site. The pharmacological properties of phytochemicals were examined in comparison to indomethacin and vitamin C, demonstrating notably remarkable anti-inflammatory effects and considerably appreciable antioxidant abilities. These findings emphasize the therapeutic potential of the compounds, including their ability to inhibit Glo-I, their anti-cancer activity, anti-inflammatory effects, and antioxidant capabilities. These insights are valuable for guiding future research in targeted therapeutic interventions.

- As marked selective antiproliferation affinities of tested agents in cell-based assays were highly signified. Future work should be directed on synthesizing new derivatives with focus on structural betterments and augmentations further elucidating the structure-activity relationship of their remarkable pharmacologies.

- Possible high throughput screenings of these promising naturals and new derivatives on other malignancy cell lines is certainly advised.

- Suggestively investigating possible molecular action mechanisms of potent natural agents (proapoptogenesis, angiogenesis, anti-platelet activity vs. the appropriately selected and respective reference agents) is warranted.

- Downstream In vivo evaluation of our derivatives for efficacy and safety cannot be overemphasized enough.

- Likely Combinations with antineoplastic agents can further underscore chemosensitisation propensies of potent agents

## Supporting information

**S1 File. Supporting data of enzyme inhibition assays, including Tables 1–16 for compounds (4-hydroxyestradiol, Shikonin, Tolcapone, Quinalizarin, Hispidin, Trichostatin A, Calceolarioside A, and Myricetin).**
(DOCX)

## Author contributions

**Conceptualization:** MohammedBashar Al-Qazzan, Qosay Al-Balas, Belal Alnajjar.

**Data curation:** MohammedBashar Al-Qazzan, Violet Kasabri, Belal Alnajjar.

**Formal analysis:** MohammedBashar Al-Qazzan, Qosay Al-Balas, Violet Kasabri.

**Investigation:** MohammedBashar Al-Qazzan, Qosay Al-Balas, Yusuf Al-Hiari.

**Methodology:** MohammedBashar Al-Qazzan, Rema AlKhateeb, Mohammed Al-Akeedi.

**Project administration:** MohammedBashar Al-Qazzan, Rema AlKhateeb.

**Resources:** Qosay Al-Balas, Violet Kasabri, Aref Zayed, Belal Alnajjar.

**Supervision:** Qosay Al-Balas, Violet Kasabri, Belal Alnajjar.

**Validation:** Yusuf Al-Hiari, Aref Zayed, Belal Alnajjar.

**Visualization:** MohammedBashar Al-Qazzan.

**Writing – original draft:** MohammedBashar Al-Qazzan, Mohammed Al-Akeedi.

**Writing – review & editing:** Qosay Al-Balas, Violet Kasabri, Yusuf Al-Hiari, Belal Alnajjar.

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
