## [Decision Letter · Decision Letter 0]

11 Jun 2025

PONE-D-25-10069Cytotoxic, Anti-inflammatory, Antioxidant, and Anti-Glyoxalase-I Evaluation of Chelating Substances; in silico and in vitro StudyPLOS ONE

Dear Dr. Al-Qazzan,

Thank you for submitting your manuscript to PLOS ONE. After careful consideration, we feel that it has merit but does not fully meet PLOS ONE’s publication criteria as it currently stands. Therefore, we invite you to submit a revised version of the manuscript that addresses the points raised during the review process.

We look forward to receiving your revised manuscript.

Kind regards,

Afzal Basha Shaik, Ph.D

Academic Editor

PLOS ONE

Comments from PLOS Editorial Office:

We note that one or more reviewers has recommended that you cite specific previously published works. As always, we recommend that you please review and evaluate the requested works to determine whether they are relevant and should be cited. It is not a requirement to cite these works. We appreciate your attention to this request.

Reviewers' comments:

Reviewer's Responses to Questions

**Comments to the Author**

1. Is the manuscript technically sound, and do the data support the conclusions?

Reviewer #1: Yes

Reviewer #2: Yes

2. Has the statistical analysis been performed appropriately and rigorously? 

Reviewer #1: Yes

Reviewer #2: Yes

3. Have the authors made all data underlying the findings in their manuscript fully available?

Reviewer #1: Yes

Reviewer #2: Yes

4. Is the manuscript presented in an intelligible fashion and written in standard English?

Reviewer #1: Yes

Reviewer #2: Yes

5. Review Comments to the Author

Reviewer #1: New relevant references should be added; kindly add the below references

1.Wan, H., Zhou, S., Li, C., Zhou, H., Wan, H., Yang, J.,... Yu, L. (2024). Ant colony algorithm-enabled back propagation neural network and response surface methodology based ultrasonic optimization of safflower seed alkaloid extraction and antioxidant. Industrial Crops and Products, 220, 119191. doi: https://doi.org/10.1016/j.indcrop.2024.119191

2.Zeng, Q., Chen, C., Chen, C., Song, H., Li, M., Yan, J.,... Lv, X. (2023). Serum Raman spectroscopy combined with convolutional neural network for rapid diagnosis of HER2-positive and triple-negative breast cancer. Spectrochimica Acta Part A: Molecular and Biomolecular Spectroscopy, 286, 122000. doi: https://doi.org/10.1016/j.saa.2022.122000

3.Ma, X., Cheng, H., Hou, J., Jia, Z., Wu, G., Lü, X.,... Chen, C. (2020). Detection of breast cancer based on novel porous silicon Bragg reflector surface-enhanced Raman spectroscopy-active structure. Chinese Optics Letters, 18(5), 051701. doi: 10.3788/COL202018.051701

4.Guo, Y., Han, Z., Zhang, J., Lu, Y., Li, C.,... Liu, G. (2024). Development of a high-speed and ultrasensitive UV/Vis-CM for detecting total triterpenes in traditional Chinese medicine and its application. Heliyon, 10(11). doi: 10.1016/j.heliyon.2024.e32239

5.Lodi, R. S., Dong, X., Wang, X., Han, Y., Liang, X., Peng, C.,... Peng, L. (2025). Current research on the medical importance of Trametes species. Fungal Biology Reviews, 51, 100413. doi: https://doi.org/10.1016/j.fbr.2025.100413

Reviewer #2: Reviewer Comments:

Manuscript Number PONE-D-25-10069

Title: Cytotoxic, Anti-inflammatory, Antioxidant, and Anti-Glyoxalase-I Evaluation of Chelating Substances; in silico and in vitro Study

The manuscript presents a comprehensive study aimed at evaluating the cytotoxic, anti-inflammatory, antioxidant, and anti-glyoxalase-I properties of chelating substances through both in silico and in vitro methodologies. This dual approach is commendable and provides a robust framework for understanding the multifaceted roles these compounds may play in therapeutic applications. The objectives are clear and align well with the significant scientific interest in chelating agents and their potential health benefits.

Strengths:

1. Innovative Approach: The integration of in silico and in vitro studies offers a powerful platform for validating findings and enhances the reliability of the results. The computational modeling adds depth to the understanding of molecular interactions and potential mechanisms of action.

2. Well-Defined Methodology: The experimental design is clearly articulated, with appropriate controls and replicates. The selection of assays for assessing cytotoxicity, inflammation, and oxidative stress is appropriate and well-justified.

3. Statistical Analysis: The use of statistical methods to analyze and interpret the data is commendable. This adds rigor to the findings and provides a baseline for future studies.

4. Relevance of Findings: The study addresses critical health-related issues, including inflammation and oxidative stress. The exploration of anti-glyoxalase-I activity is particularly relevant, given the implications for various metabolic disorders.

Areas for Improvement:

1. Literature Review: While the manuscript cites relevant studies, a more thorough discussion of the existing literature would strengthen the background. Highlighting how this study builds upon or contrasts with previous work could provide greater context for your findings.

2. In Silico Methods: The methodologies employed in the in silico portion of the study should be described in greater detail. Specific algorithms used for molecular docking and the criteria for evaluating binding affinities should be included to allow replication and validation by other researchers

3. Data Presentation: Figures and tables are pivotal for clarity. Enhancing the quality of the figures, particularly those depicting dose-response curves and molecular structures, would improve the readability of the results. Consider including error bars in your graphical data to represent variability.

4. Discussion of Mechanisms: The discussion would benefit from a deeper exploration of the mechanisms behind the observed anti-inflammatory and antioxidant effects. Speculative mechanisms could be suggested based on existing literature, helping to position the findings within the broader context of biochemical pathways

5. Conclusions and Future Directions: The conclusion section could be expanded to outline potential future research avenues that could stem from this work. Discussing the implications of the findings for clinical applications would be beneficial as well.

Minor Comments:

- Check for typographical errors and ensure consistency in terminology throughout the manuscript.

- Ensure that all references are formatted according to the journal's guidelines.

- Clarify any abbreviations at first mention in the text to maintain reader accessibility.

Overall Evaluation:

This manuscript is a valuable contribution to the field, with important implications for the design of therapeutic agents based on chelating substances. Addressing the comments outlined above will strengthen the manuscript significantly and enhance its impact in the scientific community. I recommend a minor revision before considering publication.

If possible for authors kindly cite below papers in the manuscript:

1. Cheng, Y., Wang, L., Zhang, S., Jian, W., Zeng, B., Liang, L.,... Deng, Z. (2024). The Investigation of Nfκb Inhibitors to Block Cell Proliferation in OSCC Cells Lines. Current Medicinal Chemistry. doi: 10.2174/0109298673309489240816063313

2. Li, R., Luo, P., Guo, Y., He, Y., & Wang, C. (2024). Clinical features, treatment, and prognosis of SGLT2 inhibitors induced acute pancreatitis. Expert Opinion on Drug Safety, 1-5. doi: https://doi.org/10.1080/14740338.2024.2396387

3. Kang, L., Gao, X., Liu, H., Men, X., Wu, H., Cui, P.,... Yan, J. (2018). Structure–activity relationship investigation of coumarin–chalcone hybrids with diverse side-chains as acetylcholinesterase and butyrylcholinesterase inhibitors. Molecular Diversity, 22(4), 893-906. doi: 10.1007/s11030-018-9839-y

4. Shi, S., Li, K., Peng, J., Li, J., Luo, L., Liu, M.,... Cai, W. (2022). Chemical characterization of extracts of leaves of Kadsua coccinea (Lem.) A.C. Sm. by UHPLC-Q-Exactive Orbitrap Mass spectrometry and assessment of their antioxidant and anti-inflammatory activities. Biomedicine & Pharmacotherapy, 149, 112828. doi: https://doi.org/10.1016/j.biopha.2022.112828

5. Zhou, J., Guo, Z., Peng, X., Wu, B., Meng, Q., Lu, X.,... Guo, T. (2025). Chrysotoxine regulates ferroptosis and the PI3K/AKT/mTOR pathway to prevent cervical cancer. Journal of Ethnopharmacology, 338, 119126. doi: https://doi.org/10.1016/j.jep.2024.119126

6. PLOS authors have the option to publish the peer review history of their article (what does this mean? ). If published, this will include your full peer review and any attached files.

**Do you want your identity to be public for this peer review?** For information about this choice, including consent withdrawal, please see our Privacy Policy .

Reviewer #1: No

Reviewer #2: No

---

## [Author Response · Author response to Decision Letter 1]

22 Aug 2025

Fwd: PLOS ONE Decision: Revision required [PONE-D-25-10069] - [EMID:43c9e9672f5a6941]

Inbox

mohammed Bashar

AttachmentsWed, Jun 11, 8:15 AM (6 days ago)

to me

---------- Forwarded message ---------

From: PLOS ONE <em@editorialmanager.com>

Date: Wed, 11 Jun 2025 at 8:12 AM

Subject: PLOS ONE Decision: Revision required [PONE-D-25-10069] - [EMID:43c9e9672f5a6941]

To: MohammedBashar Al-Qazzan <alanim050@gmail.com>

PONE-D-25-10069

Cytotoxic, Anti-inflammatory, Antioxidant, and Anti-Glyoxalase-I Evaluation of Chelating Substances; in silico and in vitro Study

PLOS ONE

Dear Dr. Al-Qazzan,

Thank you for submitting your manuscript to PLOS ONE. After careful consideration, we feel that it has merit but does not fully meet PLOS ONE’s publication criteria as it currently stands. Therefore, we invite you to submit a revised version of the manuscript that addresses the points raised during the review process.

A rebuttal letter that responds to each point raised by the academic editor and reviewer(s). You should upload this letter as a separate file labeled 'Response to Reviewers'. DONE

A marked-up copy of your manuscript that highlights changes made to the original version. You should upload this as a separate file labeled 'Revised Manuscript with Track Changes'. DONE

An unmarked version of your revised paper without tracked changes. You should upload this as a separate file labeled 'Manuscript'. DONE

We look forward to receiving your revised manuscript.

Kind regards,

Afzal Basha Shaik, Ph.D

Academic Editor

PLOS ONE

1. Please ensure that your manuscript meets PLOS ONE's style requirements, including those for file naming. The PLOS ONE style templates can be found at DONE

2. We note that your Data Availability Statement is currently CHANGED as follows: DONE

IT WAS THEN [All relevant data are within the manuscript and its Supporting Information files.]

NOW UPDATED TO [no some restrictions will apply] as we herein state that we have all results relevant to this work being genuinely reported in this revised version of manuscript.

EXPLANATION PROVIDED IS ["All relevant data are within the manuscript and its Supporting Information files."]

Dear Professor EIC and reviewers,

For the sake of extra clarification; you might kindly even check the ‘’ NT** of Calceolarioside A (not tested) as reported in Tables 2-4. That was unfortunately and simply THE CASE because we completely ran out of this reagent by the time we ran out of both time as well as resources for procuring ANY extra gms. If this incident can intervene with our chances of considering our draft for PLoS One approval and acceptance; we can omit this agent from Tables 2-4! Kindly please advice! Honestly we have no further disclosures to make at this stage of our comprehensive bulk of reported results.

For example, authors should submit the following data: N/A [based to above data availability statement]

Comments from PLOS Editorial Office:

We note that one or more reviewers has recommended that you cite specific previously published works. As always, we recommend that you please review and evaluate the requested works to determine whether they are relevant and should be cited. It is not a requirement to cite these works. We appreciate your attention to this request. DONE. Kindly check modified text of both introduction and discussion parts.

Reviewers' comments:

Reviewer's Responses to Questions

Comments to the Author

1. Is the manuscript technically sound, and do the data support the conclusions?

Reviewer #1: Yes

Reviewer #2: Yes

2. Has the statistical analysis been performed appropriately and rigorously?

Reviewer #1: Yes

Reviewer #2: Yes

3. Have the authors made all data underlying the findings in their manuscript fully available?

The PLOS Data policy requires authors to make all data underlying the findings described in their manuscript fully available without restriction, with rare exception (please refer to the Data Availability Statement in the manuscript PDF file). The data should be provided as part of the manuscript or its supporting information, or deposited to a public repository. For example, in addition to summary statistics, the data points behind means, medians and variance measures should be available. If there are restrictions on publicly sharing data—e.g. participant privacy or use of data from a third party—those must be specified. N/A

Reviewer #1: Yes

Reviewer #2: Yes

Our statement as above mentioned and explained:

[‘’ no some restrictions will apply ‘’ and ‘’All relevant data are within the manuscript and its Supporting Information files.’’]

4. Is the manuscript presented in an intelligible fashion and written in standard English?

Reviewer #1: Yes

Reviewer #2: Yes

5. Review Comments to the Author

Reviewer #1: New relevant references should be added; kindly add the below references DONE as NEWLY INSERTED reference 82; KINDLY SEE BELOW responses:

Exceptionally these 2 references (Zeng et al., 2023; and Ma et al.,2020) could not fit or match any core concept(s) within our draft (we do not touch base with any malignancies’ advanced technologies of diagnostics, for our deep dismay and disappointment!)

a. Zeng Q, Che, C, Chen C, Song H, Li M, Yan J, Lv X. Serum Raman spectroscopy combined with convolutional neural network for rapid diagnosis of HER2-positive and triple-negative breast cancer. Spectrochimica Acta Part A: Molecular and Biomolecular Spectroscopy 2023; 286:122000. https://doi.org/ 10.1016/j.saa.2022.122000

b. Ma X, Cheng H, Hou J, Jia Z, Wu G, Lü X, Chen C. Detection of breast cancer based on novel porous silicon Bragg reflector surface-enhanced Raman spectroscopy-active structure. Chinese Optics Lett. 2020; 18(5):051701. https://doi.org/10.3788/COL202018.051701

………………………………………………………………………………………………………………………Reviewer #2: Reviewer Comments:

Manuscript Number PONE-D-25-10069

Title: Cytotoxic, Anti-inflammatory, Antioxidant, and Anti-Glyoxalase-I Evaluation of Chelating Substances; in silico and in vitro Study

The manuscript presents a comprehensive study aimed at evaluating the cytotoxic, anti-inflammatory, antioxidant, and anti-glyoxalase-I properties of chelating substances through both in silico and in vitro methodologies. This dual approach is commendable and provides a robust framework for understanding the multifaceted roles these compounds may play in therapeutic applications. The objectives are clear and align well with the significant scientific interest in chelating agents and their potential health benefits.

Strengths:

1. Innovative Approach: The integration of in silico and in vitro studies offers a powerful platform for validating findings and enhances the reliability of the results. The computational modeling adds depth to the understanding of molecular interactions and potential mechanisms of action.

2. Well-Defined Methodology: The experimental design is clearly articulated, with appropriate controls and replicates. The selection of assays for assessing cytotoxicity, inflammation, and oxidative stress is appropriate and well-justified.

3. Statistical Analysis: The use of statistical methods to analyze and interpret the data is commendable. This adds rigor to the findings and provides a baseline for future studies.

4. Relevance of Findings: The study addresses critical health-related issues, including inflammation and oxidative stress. The exploration of anti-glyoxalase-I activity is particularly relevant, given the implications for various metabolic disorders.

Areas for Improvement:

1. Literature Review: While the manuscript cites relevant studies, a more thorough discussion of the existing literature would strengthen the background. Highlighting how this study builds upon or contrasts with previous work could provide greater context for your findings. DONE [into introduction; 10 newly added references were inserted into [18-21] sequence for absolute relevance of promising therapeutic implementations of glyoxylase interventions]

2. In Silico Methods: The methodologies employed in the in silico portion of the study should be described in greater detail. Specific algorithms used for molecular docking and the criteria for evaluating binding affinities should be included to allow replication and validation by other researchers. DONE. [Please see added text in respective subheading].

3. Data Presentation: Figures and tables are pivotal for clarity. Enhancing the quality of the figures, particularly those depicting dose-response curves and molecular structures, would improve the readability of the results. Consider including error bars in your graphical data to represent variability. DONE. [Please see enhanced fig.s of Table 1].

4. Discussion of Mechanisms: The discussion would benefit from a deeper exploration of the mechanisms behind the observed anti-inflammatory and antioxidant effects. Speculative mechanisms could be suggested based on existing literature, helping to position the findings within the broader context of biochemical pathways. DONE. [in Speculative molecular action mechanisms of antitumor -antiinflammation cross talks biology: Comprehensively included with supportive references 84-95].

5. Conclusions and Future Directions: The conclusion section could be expanded to outline potential future research avenues that could stem from this work. Discussing the implications of the findings for clinical applications would be beneficial as well. DONE. [in Concluding remarks and future directives]

Minor Comments:

- Check for typographical errors and ensure consistency in terminology throughout the manuscript. DONE

- Ensure that all references are formatted according to the journal's guidelines. DONE

- Clarify any abbreviations at first mention in the text to maintain reader accessibility. DONE

Overall Evaluation:

This manuscript is a valuable contribution to the field, with important implications for the design of therapeutic agents based on chelating substances. Addressing the comments outlined above will strengthen the manuscript significantly and enhance its impact in the scientific community. I recommend a minor revision before considering publication.

If possible for authors kindly cite below papers in the manuscript: DONE as reference 82:

c. Cheng Y, Wang L, Zhang S, Jian W, Zeng B, Liang L, Deng Z. The Investigation of Nfκb Inhibitors to Block Cell Proliferation in OSCC Cells Lines. Curr Med Chem. 2024. https://doi.org/10.2174/ 0109298673309489240 816063313

d. Li R, Luo P, Guo Y, He Y, Wang C. Clinical features, treatment, and prognosis of SGLT2 inhibitors induced acute pancreatitis. Expert Opin Drug Saf. 2024; 4:1-5. https://doi.org/10.1080/14740338.2024.2396387

e. Kang L, Gao XH, Liu HR, Men X, Wu HN, Cui PW, Oldfield E, Yan JY. Structure-activity relationship investigation of coumarin-chalcone hybrids with diverse side-chains as acetylcholinesterase and butyrylcholinesterase inhibitors. Mol Divers. 2018; 22(4):893-906. https://doi.org/10.1007/s11030-018-9839-y

f. Shi S, Li K, Peng J, Li J, Luo L, Liu M, Chen Y, Xiang Z, Xiong P, Liu L, Cai W. Chemical characterization of extracts of leaves of Kadsua coccinea (Lem.) A.C. Sm. by UHPLC-Q-Exactive Orbitrap Mass spectrometry and assessment of their antioxidant and anti-inflammatory activities. Biomed Pharmacother. 2022; 149:112828. https://doi.org/10.1016/j.biopha. 2022.112828

g. Zhou J, Guo Z, Peng X, Wu B, Meng Q, Lu X, Feng L, Guo T. Chrysotoxine regulates ferroptosis and the PI3K/AKT/mTOR pathway to prevent cervical cancer. J Ethnopharmacol. 2025; 338(Pt3):119126. https://doi.org/10.1016/j.jep. 2024.119126

h. Lodi RS, Dong X, Wang X, Han Y, Liang X, Peng C, Peng L. Current research on the medical importance of Trametes species. Fung Biol Rev. 2025; 51:100413. https://doi.org/10.1016/j.fbr.2025.100413

i. Guo Y, Han Z, Zhang J, Lu Y, Li C, Liu G. Development of a high-speed and ultrasensitive UV/Vis-CM for detecting total triterpenes in traditional Chinese medicine and its application. Heliyon, 2024; 10(11). https://doi.org/10.1016/j.heliyon.2024.e32239

j. Wan H,

---

## [Editor Report · Decision Letter 1]

15 Sep 2025

Cytotoxic, Anti-inflammatory, Antioxidant, and Anti-Glyoxalase-I Evaluation of Chelating Substances; in silico and in vitro Study

PONE-D-25-10069R1

Dear Dr. Al-Qazzan,

We’re pleased to inform you that your manuscript has been judged scientifically suitable for publication and will be formally accepted for publication once it meets all outstanding technical requirements.

Kind regards,

Afzal Basha Shaik, Ph.D

Academic Editor

PLOS ONE
---

## [Editor Report · Acceptance letter]

PONE-D-25-10069R1

PLOS ONE

Dear Dr. Al-Qazzan,

I'm pleased to inform you that your manuscript has been deemed suitable for publication in PLOS ONE. Congratulations! Your manuscript is now being handed over to our production team.

Kind regards,

on behalf of

Dr. Afzal Basha Shaik

Academic Editor

PLOS ONE